# Modulating electrospun polycaprolactone scaffold morphology and composition to alter endothelial cell proliferation and angiogenic gene response

**James Alexander Reid, Alison McDonald, Anthony Callanan** *

Institute for Bioengineering, School of Engineering, The University of Edinburgh, Edinburgh, United Kingdom

* Anthony.Callanan@ed.ac.uk

**Data Availability Statement:** All relevant data are within the manuscript and its Supporting Information files.

## Abstract

The aim of this study was to look at how the composition and morphology of polymer scaffolds could be altered to create an optimized environment for endothelial cells. Four polycaprolactone (PCL) scaffolds were electrospun with increasing fibre diameters ranging from 1.64 μm to 4.83 μm. The scaffolds were seeded with human umbilical vein endothelial cells (HUVEC) and cultured for 12 days. PCL scaffolds were then electrospun incorporating decellularized bovine aorta ECM and cultured in a hypoxic environment. We noted deeper cell infiltration on the largest fibre diameter compared to the other three scaffolds which resulted in an increase in the gene expression of CD31; a key angiogenic marker. Increased cell viability and cell proliferation were also noted on the largest fibre. Furthermore, we noted that the incorporation of extracellular matrix (ECM) had minimal effect on cell viability, both in normoxic and hypoxic culture conditions. Our results showed that these environments had limited influences on hypoxic gene expression. Interestingly, the major findings from this study was the production of excretory ECM components as shown in the scanning electron microscopy (SEM) images. The results from this study suggest that fibre diameter had a bigger impact on the seeded HUVECs than the incorporation of ECM or the culture conditions. The largest fibre dimeter (4.83 μm) is more suitable for seeding of HUVECs.

## Introduction

Currently, vascular diseases such as coronary heart disease and strokes account for upwards of 30% of all deaths in Europe [1]. There are a wide array of strategies being investigated to come up with novel treatment methods for vascular disease. These include mechanical cues, biochemical cues (incorporation of native ECM (extracellular matrix), protein binding, etc.) and bioelectric cues (electrical stimulation) [2–11]. Electrospinning is an exciting avenue that allows for a wide range of scaffold morphologies to be created [12–15]. This method has been used in many aspects of tissue engineering to mimic the structure of the native ECM, allowing for improved environments for cell adhesion, migration and proliferation [16, 17]. Polycaprolactone (PCL) is one of the most prominently used polymers for electrospinning, due to its widely studied biocompatibility and its ease of manufacturing [18–20].

**Funding:** The rewards were received by James Reid and Anthony Callanan. This work was funded by an Engineering & Physical Sciences Research Council (ESPRC) doctoral training partnership studentship EP/N509644/1 (https://epsrc.ukri.org/skills/students/dta/) and a UK Regenerative Medicine Platform II grant MR/L022974/1 (https://www.ukrmp.org.uk/) The funders had no role in study design, data collection and analysis, decision to publish, or preparation of the manuscript.

**Competing interests:** The authors have declared that no competing interests exist.

To treat this, approaches such as bypass grafting are implemented [21]. This surgery is often used in the treatment of coronary heart disease (blockage in the coronary artery) and peripheral arterial disease (blockage in arteries in the arms and legs). This is either done with a synthetic material, such as PTFE, or by using an autologous vessel such as the saphenous vein or the internal thoracic artery [3]. These autologous vessels represent the gold standard for bypass grafting, especially for smaller vessel bypasses ($< 6$ mm), as they are associated with much higher patency rates compared to their synthetic counterparts (80–90% for the saphenous vein after 5 years compared to 20–70% for the PTFE conduit) [3]. However, there are only a finite amount of vessels that can be harvested for bypass grafting, and patients who are in need of a bypass graft will often have damaged autologous vessels that are not of the standard required for surgery [22]. Therefore, there is a clear need for an easily manufacturable synthetic alternative that can achieve higher patency rates.

One of the major aims when developing new treatments for vascular disease is to create a healthy layer of endothelial cells organized into a tubular network [23]. This can happen under the correct environmental conditions, whereby endothelial cells communicate with each other through the release of angiogenic paracrine factors [24–26]. One way of doing this is by providing the correct physical and biological cues to the cells through altering the topographical features of the scaffold and incorporating native ECMs [9]. For example, it has been shown with human kidney primary epithelial cells and chondrocytes that fibre diameter and fibre orientation have dramatic effects on cell morphology and the expression of key genes [27, 28]. While there has been some work looking at fibre diameter and endothelial cells, to the best of our knowledge, the present study is the first that looks at the effect of fibre diameter in electrospun PCL scaffolds on endothelial cells [29, 30]. Furthermore, recent work by our group has shown that incorporating native vascular ECMs into the fibre had positive effects on HUVEC proliferation and gene expression [9, 31]. While these ECM scaffolds have been looked at in normoxic culture, they have not yet been studied in hypoxic cultures that more accurately mimic the native oxygen content found in most native tissue types. Therefore, there is further motivation in not only studying fibre diameter but also looking at how hypoxic culture affects blended PCL/ECM electrospun scaffolds. By looking at these different compositional and morphological aspects of the electrospun scaffold, the aim is to pin-point which one has the biggest effect on seeded endothelial cells to help guide future scaffold design.

Herein, we propose altering the fibre size of PCL electrospun scaffolds and incorporating aortic ECM into the fibres to study how these aspects of the scaffold affect endothelial cell performance in both normoxic and hypoxic conditions.

## Materials and methods

### Electrospinning

Polycaprolactone (Mn = 80,000 Da, Sigma Aldrich) was either dissolved into Hexafluoroisopropanol (HFIP) (Manchester Organics) or a 5:1 mixture of Chloroform:Methanol (C:M) (both from Sigma Aldrich). Electrospun fibres were collected onto aluminium foil (Fig 1) on an 8 cm diameter rotating mandrel, as per a previously described protocol [31].

### Scanning electron microscopy

Unseeded scaffolds were imaged using a Hitachi S4700 fuelled emission scanning electron microscope (SEM, Hitachi) with a 5 kV accelerating voltage and a 12 mm working distance. All scaffolds were sputter coated prior to imaging with an Emscope SC500A splutter coating using gold-palladium at a ratio of 60:40. Seeded scaffolds were visualized using an Osmium Tetroxide staining technique [32].

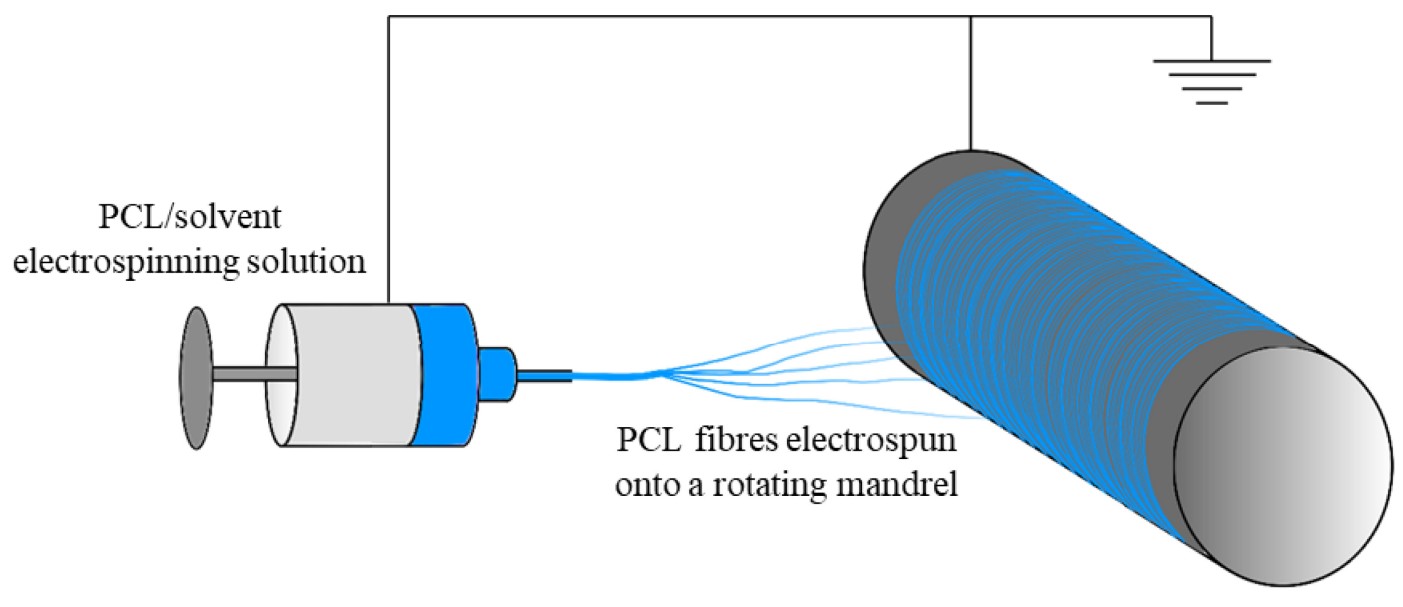

**Fig 1. Electrospinning technique.**

### Fibre and pore properties

Scanning electron images were analysed using ImageJ software (NIH). Briefly, SEM images of the scaffolds analysed using the DiameterJ plugin for fibre diameter and pore width and the OrientationJ plugin for fibre orientation [33]. Fibre diameters were measured on a minimum for 50 different fibres on four different scaffolds. Variance in fibre diameters along individual fibres was measured using 5 diameter measurements per fibre along a total of 5 fibres.

### Mechanical testing

Tensile properties were measured using an Instron 3367 testing machine (Instron) with a 50 N load cell. Briefly, 40 mm x 5 mm strips of scaffold (n = 5) were cut and were stretched with a starting gauge length of 20 mm. Scaffolds were stretched to failure at 10 mm/min. Incremental Young's moduli were calculated using the formula:

$$E_{incremental} = \frac{\Delta\sigma}{\Delta\varepsilon} = \frac{FL_0}{A\Delta L}$$

where $E_{incremental}$ is Young's Modulus between two strain bands, σ is stress, ε is strain, $F$ is the applied force, $A$ is the cross sectional area, $\Delta L$ is change in length and $L_0$ is the original length. Young's Moduli for each scaffold were analysed and expressed at regular incremental intervals, as previously described [34–37].

The thickness of each scaffold was measured using a DMK 41AU02 monochrome camera in order to deduce the cross sectional area (n = 5). Analysis of the scaffolds was done on ImageJ (NIH).

### Contact angle measurement

Contact angle was measured on dry scaffolds (n = 5). A 5 µl droplet of water was placed onto the scaffold and images were captured using a DMK 41AU02 monochrome camera at a frequency of 5 Hz. Analysis of the scaffolds was done on ImageJ (NIH) using the LBADSA plugin [38].

## Scaffold porosity

Scaffold porosity (%) was calculated (n = 5) using the following equation:

$$Porosity = 100 \times \left( 1 - \frac{V_{scaffold} \big/ m_{scaffold}}{\rho_{polymer}} \right)$$

where $V$ is volume, $m$ is mass and $\rho$ is density. The thickness of the scaffold was measured in order to calculate the volume of the scaffold. This was done using a DMK 41AU02 monochrome camera. All scaffolds were punched out using a 10 mm diameter punch.

## Cell culture

Human umbilical vein endothelial cells (HUVECs) from an infant male Caucasian donor were obtained cryopreserved at passage 1 (Pro-moCell GmbH) and expanded to passage 7 in a 5% $CO_2$/37˚C atmosphere. This study abides by all criteria of the UK Human Tissue Act. HUVECs were expanded using MCBD 131 medium (Life Technologies) supplemented with 5% v/v fetal bovine serum (FBS; ThermoFisher Scientific); 1% v/v L-glutamine; 1% v/v penicillin/streptomycin (Life Technologies); 1 mg/L hydrocortisone; 50 mg/L ascorbic acid (Sigma); 2 µg/L fibroblast growth factor (PeproTech); 10 µg/L epidermal growth factor (PeproTech); 2 µg/L insulin-like growth factor (PeproTech); and 1 µg/L vascular endothelial growth factor (PeproTech). All cells were subsequently cultured with this complete media.

## CellTiter-Blue® cell viability assay

The assay was performed as per manufacturer's instructions (Promega) and as described in previous studies [9]. Measurements were taken after 3.5 h at Ex: 525 nm and Em: 580–640 nm. For each condition group, n = 4. All data has been represented with the background fluorescence removed (negative control). A comparison of cells cultured on scaffolds to cells cultured on tissue culture plastic can be found in S1 Fig.

## Cell staining

Scaffolds used for cell staining were stained in 0.1% v/v 1000X Phalloidin-iFluor™514 conjugate (AAT Bioquest, Stratech) and 300 nM 4',6-diamidino-2-phenylindole (DAPI) (Sigma-Aldrich, UK) as previously described [9].

## Cell imaging

Cell seeded scaffolds were visualized using a custom-built multiphoton microscope previously reported [39]. Coherent anti-stokes Raman scattering (CARS) imaging at 2911 cm⁻¹ was used to image the PCL scaffold fibres, whilst simultaneously exciting two photon fluorescence (TPEF) from Phalloidin and DAPI stained cells. All images were acquired using a 25x/1.05 N. A water immersion lens (XLPlanN, Olympus), providing a maximum field of view of 509 µm on the sample. To measure infiltration of cells into the scaffold Z-stacks were acquired with 1 µm steps.

Cell seeded scaffolds were also visualized using SEM using a previously described osmium based method [40]. These osmium stained scaffolds were visualized using a Hitachi TM4000 tabletop SEM (Hitachi) with a 15 kV accelerating voltage and a 10mm working distance. These scaffolds were sputter coated with an Elmscope FLM-007 using gold.

## Measuring cell infiltration

Cell infiltration was measured using DAPI and phalloidin Z-stack images. Images were imported into imageJ (NIH) whereby infiltration depth was measured.

## Reverse transcription polymerase chain reaction (RT-PCR)

RNA was extracted from the cell seeded scaffolds using a Tri-Reagent (Invitrogen, Thermofisher) method and purified using Qiagen's RNeasy spin column system. Real-time polymerase chain reaction was performed using a LightCycler® 480 Instrument II (Roche Life Science) and Sensifast™ SYBR® High-ROX system (Bioline). Forward and reverse sequences were designed online. Relative quantification of RT-PCR results was carried out using the $2^{-\Delta\Delta ct}$ method [41]. Gene expression levels were expressed relative to GAPDH (housekeeping gene) and normalised to 70% confluent HUVECs on tissue culture plastic (positive control).

## Statistical analysis

Data was expressed as mean ± 1 standard deviation. Statistical analysis was performed using one-way ANOVA with post-hoc Tukey test, where $^*p<0.05$, $^{**}p<0.01$ and $^{***}p<0.001$.

## The effect of fibre diameter on endothelial cells

**Electrospinning.** PCL was either dissolved into HFIP or a 5:1 mixture of Chloroform: Methanol (C:M at varying concentrations to achieve four different solutions for electrospinning. The four different fibre sizes were achieved by altering a variety of parameters all listed in Table 1.

**Timepoints.** Timepoints of 1 day, 6 days and 12 days were used in the fibre diameter studies. These timepoints allow for sufficient time to assess these scaffolds under conventional *in vitro* culture conditions.

**Scaffold seeding and culture.** HUVECs were lifted for scaffold seeding at 80% confluence. Briefly, 20,000 cells/cm$^2$ were drip seeded onto 10mm diameter scaffolds, as per a previously described seeding method [9]. The effects of fibre diameter were assessed at timepoints of 24 hours, 6 days and 12 days.

**Genes analysed using RT-PCR.** Forward and reverse primers used in the fibre diameter experiments are displayed in Table 2.

## The effects of incorporating ECM into polymer fibres and culture conditions on endothelial cells

**Aorta decellularization.** Bovine aorta ECM was harvested and decellularized using a previously described protocol [9, 31]. Briefly, aortic ECM was harvested from a 2 year old cow, obtained post-mortem from a local slaughterhouse. This study abides by all criteria of the UK Human Tissue Act. Samples were cut into 40 mm discs and perfusion decellularized in 0.5%

**Table 1. Electrospinning parameters for fibre diameter study.**

| Fibre size | Polymer concentration (%) | Solvent used | Needle bore (mm) | Flow rate (mL/h) | Total volume (mL) | Distance between needle tip and mandrel (cm) | Positive voltage (kV) | Negative Voltage (kV) | Mandrel rotational speed (RPM) |
|---|---|---|---|---|---|---|---|---|---|
| S | 8 | HFIP | 0.4 | 1 | 6 | 11 | +10.6 | -2 | 250 |
| M | 12 | HFIP | 0.4 | 1.7 | 8.5 | 15 | +14 | 0 | 250 |
| L | 14 | 5:1 C:M | 0.8 | 3 | 18 | 19 | +15 | -4 | 250 |
| XL | 19 | 5:1 C:M | 0.8 | 4 | 20 | 23 | +18 | -4 | 250 |

**Table 2. Primer sequences used for fibre diameter RT-PCR.**

| Gene | Primer | Sequence | Reference |
|---|---|---|---|
| Glyceraldehyde 3-phosphate dehydrogenase (GAPDH) | GAPDH (forward) | GTCTCCTCTGACTTCAACAG | [23] |
| | GAPDH (reverse) | GTTGTCATACCAGGAAATGAG | |
| Matrix metalloproteinase-1 (MMP1) | MMP1 (forward) | CGGTTTTTCAAAGGGAATAAGTACT | [23] |
| | MMP1 (reverse) | TCAGAAAGAGCAGCATCGATATG | |
| matrix metalloproteinase-2 (MMP2) | MMP2 (forward) | CGCTCAGATCCGTGGTGAG | [23] |
| | MMP2 (reverse) | TGTCACGTGGCGTCACAGT | |
| Tissue inhibitor of metalloproteinases-2 (TIMP2) | TIMP2 (forward) | AATGCAGATGTAGTGATCAGG | [23] |
| | TIMP2 (reverse) | TCTATATCCTTCTCAGGCCC | |
| Vascular endothelial growth factor (VEGF) | VEGF (forward) | AGACCAAAGAAAGATAGAGCAAGACAAG | [23] |
| | VEGF (reverse) | GGCAGCGTGGTTTCTGTATCG | |
| Platelet endothelial cell adhesion molecule (CD31) | CD31 (forward) | ACTGGACAAGAAAGAGGCCATCCA | [42] |
| | CD31 (reverse) | TCCTTCTGGATGGTGAAGTTGGCT | |

w/v sodium dodecyl sulfate (SDS) for 36 h [6]. Decellularization was confirmed using a Quant-iT™ PicoGreen™ dsDNA Assay Kit (ThermoFisher, UK) and through Eosin and Haematoxylin staining as previously described [31]. The decellularized ECM was lyophilized and turned into a powder using a ball mill.

**Electrospinning for ECM incorporation study.** Powdered ECM was dissolved into hexafluoroisopropanol (HFIP) at 0.25% w/v. PCL was the added to the HFIP/ECM solution at 8% w/v and dissolved overnight using agitation. The solution was electrospun using the following parameters: 0.4 mm needle, 0.8 mL/h for 6 h, 12 cm distance to mandrel, +14 kV, −4 kV, 250 RPM. A PCL only control was electrospun alongside using the same parameters.

**Timepoints.** Timepoints of 12 hours, 24 hours and 48 hours were used in the ECM and hypoxic studies. These timepoints allow for sufficient time to assess these scaffolds under hypoxic *in vitro* culture conditions.

**Scaffold seeding and hypoxic culture.** HUVECs at P7 were lifted at 80% confluence and drip seeded at 60,000 cells/cm$^2$ on 10 mm diameter scaffolds, as per a previously described seeding protocol [9]. The effects of ECM incorporation and hypoxic culture were assessed at timepoints of 12 hours, 24 hours and 48 hours. All scaffolds were cultured in normoxic conditions (16% $O_2$/ 5% $CO_2$) for 2 hours to assist with cell binding. Scaffolds in the normoxia group were left in the normoxic incubator after the 2h period. Scaffolds in the hypoxia group were transferred over to an incubator set at 2% $O_2$ / 5% $CO_2$. The timepoints started after this 2 hour period of normoxic incubation. See section 2.12 for more detail on cell culture.

**Genes analysed using RT-PCR.** Forward and reverse primers used in the ECM and hypoxia experiments are displayed in Table 3.

## Results

### Scaffold fibre properties for electrospun scaffolds with different fibre sizes

Four different fibre diameters were successfully electrospun: 1.64 ± 0.18 μm, 2.95 ± 0.16 μm, 3.37 ± 0.27 μm and 4.83 ± 0.49 μm. Statistical significance was noted between the four fibre diameters (p < 0.001 for all cases) meaning they can all be considered as being different (Fig 2A).

Furthermore, the fibre orientation on all four scaffolds appears to be very similar with no evident peaks in frequency at any angle. This signifies that the scaffolds all have a uniform random alignment (Fig 2B). As expected, scaffold pore size increased with fibre diameter. A very

**Table 3. Primer sequences used for ECM and hypoxia RT-PCR.**

| Gene | Primer | Sequence | Reference |
|------|--------|----------|-----------|
| Glyceraldehyde 3-phosphate dehydrogenase (GAPDH) | GAPDH (forward) | GTCTCCTCTGACTTCAACAG | [23] |
| | GAPDH (reverse) | GTTGTCATACCAGGAAATGAG | |
| Basic Fibroblast Growth Factor (FGF2) | FGF2 (forward) | CACCTATAATTGGTCAAAGTGGTTG | Self-designed |
| | FGF2 (reverse) | AAACGAGGGAGAAAGGATGGA | |
| Hypoxia-inducible Factor 1-α (HIF1-α) | HIF1-α (forward) | ACTTGGCAACCTTGGATTGG | Self-designed |
| | HIF1-α (reverse) | GTGCAGTGCAATACCTTCCA | |
| Nitric Oxide Synthase 3 (eNOS) | eNOS (forward) | AGATGTTCCAGGCTACAATCCGCT | [42] |
| | eNOS (reverse) | TGTATGCCAGCACAGCTACAGTGA | |

strong linear correlation was noted between these two scaffold parameters with and $R^2$ value of 0.9997 (Fig 2C). Pore diameter ranged from 8.4 ± 3.7 μm in the small fibre scaffold up to 23.3 ± 9.0 μm in the extra-large fibre scaffold (Table 4). Furthermore, it was noted that scaffold porosity decreased as fibre diameter increased. The small fibre had a porosity of 91.0 ± 1.6%, decreasing all the way down to 83.3 ± 0.9% in the extra-large fibre scaffold (Table 4).

## Mechanical properties of scaffolds with different fibre sizes

Mechanical testing (tensile testing) showed some interesting results. Firstly, the ultimate tensile strength (UTS) of the scaffold was highest in the medium and large fibre scaffolds, but significantly dropped either side of this; both in the small fibre and extra-large fibre (p values ranging from 0.001 to 0.068) (Table 4). Similarly, the failure strain of the scaffolds was highest in the medium and large fibre scaffolds, but dropped significantly in the small and extra-large scaffolds (p values ranging from 0.001 to 0.019) (Table 4). Representative stress strain curves can be seen in Fig 2D.

The small fibre scaffold had significantly lower stiffness at both the 0–5% (p < 0.001) and 5–10% (p < 0.017) strain bands than the three other fibre diameters. Interestingly, the larger the fibre, the more stiffness it maintained through the two strain bands, conserving more elasticity as strain increased.

Contact angle measurements lead to some interesting results (Table 4). There was a general trend in increasing contact angle as fibre size increased–ranging from 110.8 ± 7.7˚ in the small fibre up to 131.5 ± 1.1˚ in the extra-large fibre. A significant difference was noted between the contact angles on the small fibre scaffold compared to the three other scaffolds (p < 0.001).

## Cell viability of HUVECs on scaffolds with different fibre diameters

Results showed that the extra-large fibre lead to significantly higher cell viability after 12 days than the three other scaffolds (p < 0.001). RFU values were approximately 200% to 300% higher than the three other scaffold morphologies (Fig 2E). Furthermore, higher viabilities were also noted in the medium, large and extra-large scaffolds after 24 hours compared to the small scaffold.

## Visualizing cells grown on scaffolds with different fibre diameters

SEM images (Fig 3) of osmium stained HUVECs on all four fibre morphologies show that the cells are spreading across the fibres and have multiple binding sites. The images also show that the cells are adopting a slightly more elongated shape when bound to the largest fibre diameter scaffold. Likewise, DAPI and phalloidin staining Z-stack images (Fig 4) showed that the larger fibres lead to the HUVECs spreading out across the fibre, adopting a more desirable elongated

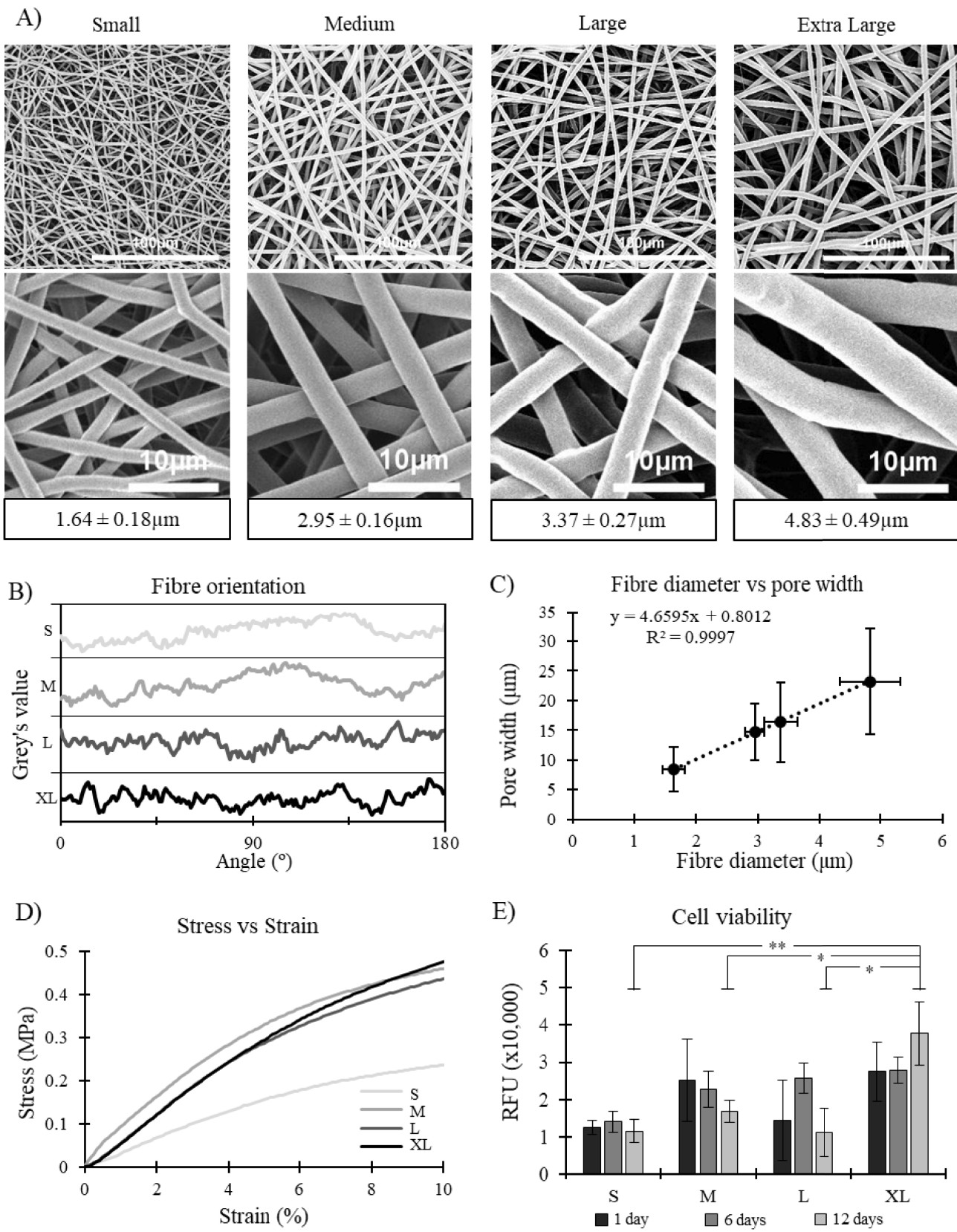

**Fig 2. A)** Representative SEM images of the four scaffolds at two different magnifications. Fibre diameters listed below. Mean ± SD. **B)** Fibre orientation of all four scaffolds, showing high degree of randomness. **C)** Correlation between fibre diameter and pore width for all four scaffolds. $R^2$ value of 0.9997 suggests an extremely high correlation. **D)** Representative stress strain curves for all four scaffolds. **E)** Cell viability for all four scaffold morphologies, RFU = relative fluorescence units, Mean ± SD, n = 4.

shape. This spreading became more evident in the later time points. In contrast, the small fibre showed a more rounded cell shape, which may indicate apoptosis [43].

Z-stack imaging also allowed for cell infiltration to be assessed (Fig 4). The larger fibre scaffold with larger pores showed infiltration up to approximately 80 μm, compared to approximately 25–30 μm of infiltration for the other three scaffolds.

## Altered gene expressions on scaffolds with different fibre diameters

Interesting trends were noted across the four scaffolds (Fig 5). Most notably, CD31 expression was significantly higher in the extra-large fibre scaffold after 6 and 12 days compared to the medium fibre diameter. While no other significance was noted in the CD31 expression, there is an evident trend with the extra-large scaffold having higher relative expression than the three other fibre sizes at both 6 days and 12 days of culture. MMP1, MMP2 and TIMP2 all showed very similar results to each other. In all cases the small fibre diameter showed a significantly lower relative expression at 12 days compared to 6 days (with MMP1 at 12 days being significantly lower than the 3 other fibre diameters at 12 days as well). The other three morphologies did show downregulations between 6 and 12 days, however, none of these were significant. No real trends were noted in VEGF relative expression across the four scaffolds.

## Osmium stained PCL and ECM scaffolds cultured in different environments visualized using SEM

The representative images of osmium stained HUVECs reveal some interesting results (Fig 6A). Osmium cell staining also stains extraneous factors such as collagen and other proteins produced by the cells during the experimental culture period. Under both normoxic and hypoxic culture, the ECM incorporated scaffolds showed increased cell coverage and extraneous factors deposited by the cells compared to the PCL scaffold. The ECM scaffold had approximately 80% and 70% scaffold coverage when cultured under normoxic and hypoxic conditions, respectively, compared to approximately 10% and 20% for the PCL scaffold.

## Cell viability and total DNA content on PCL and ECM scaffolds cultured in different environments

Cell viability and DNA quantification both showed very similar results (Fig 6B and 6C). The major differences noted were between the PCL and ECM scaffolds cultured in normoxic conditions, where the PCL scaffold showed significantly higher viability and DNA content than the ECM scaffold after 12h and 48h of culture. When cultured in hypoxic conditions, the ECM and PCL scaffolds had similar viabilities and DNA contents, with no significance noted.

**Table 4. Mechanical and physical properties of all four scaffolds.**

| | Fibre diameter (μm) | Pore diameter (μm) | Variance in fibre diameter along fibre (%) | Scaffold thickness (μm) | Porosity (%) | Ultimate tensile strength (MPa) | Failure strain (%) | Contact angle at 0.2s (˚) | Stiffness (MPa) | |
|---|---|---|---|---|---|---|---|---|---|---|
| | | | | | | | | | 0–5% | 5–10% |
| **S** | 1.64 ± 0.18 | 8.4 ± 3.7 | 1.23 | 140 ± 9.4 | 91.0 ± 1.6 | 1.08 ± 0.17 | 843 ± 143 | 110.8 ± 7.7 | 4.10 ± 0.33 | 2.17 ± 0.25 |
| **M** | 2.95 ± 0.16 | 14.7 ± 4.8 | 0.83 | 336 ± 36.9 | 89.6 ± 2.4 | 1.50 ± 0.09 | 1202 ± 62 | 128.2 ± 4.0 | 6.50 ± 0.72 | 2.99 ± 0.46 |
| **L** | 3.37 ± 0.27 | 16.4 ± 6.7 | 2.28 | 549 ± 33.8 | 83.9 ± 1.1 | 1.55 ± 0.16 | 1071 ± 21 | 129.1 ± 2.5 | 5.80 ± 0.36 | 3.16 ± 0.48 |
| **XL** | 4.83 ± 0.49 | 23.3 ± 9.0 | 2.83 | 598 ± 59.6 | 83.3 ± 0.9 | 1.26 ± 0.06 | 855 ± 120 | 131.5 ± 1.1 | 5.77 ± 0.39 | 3.66 ± 0.28 |

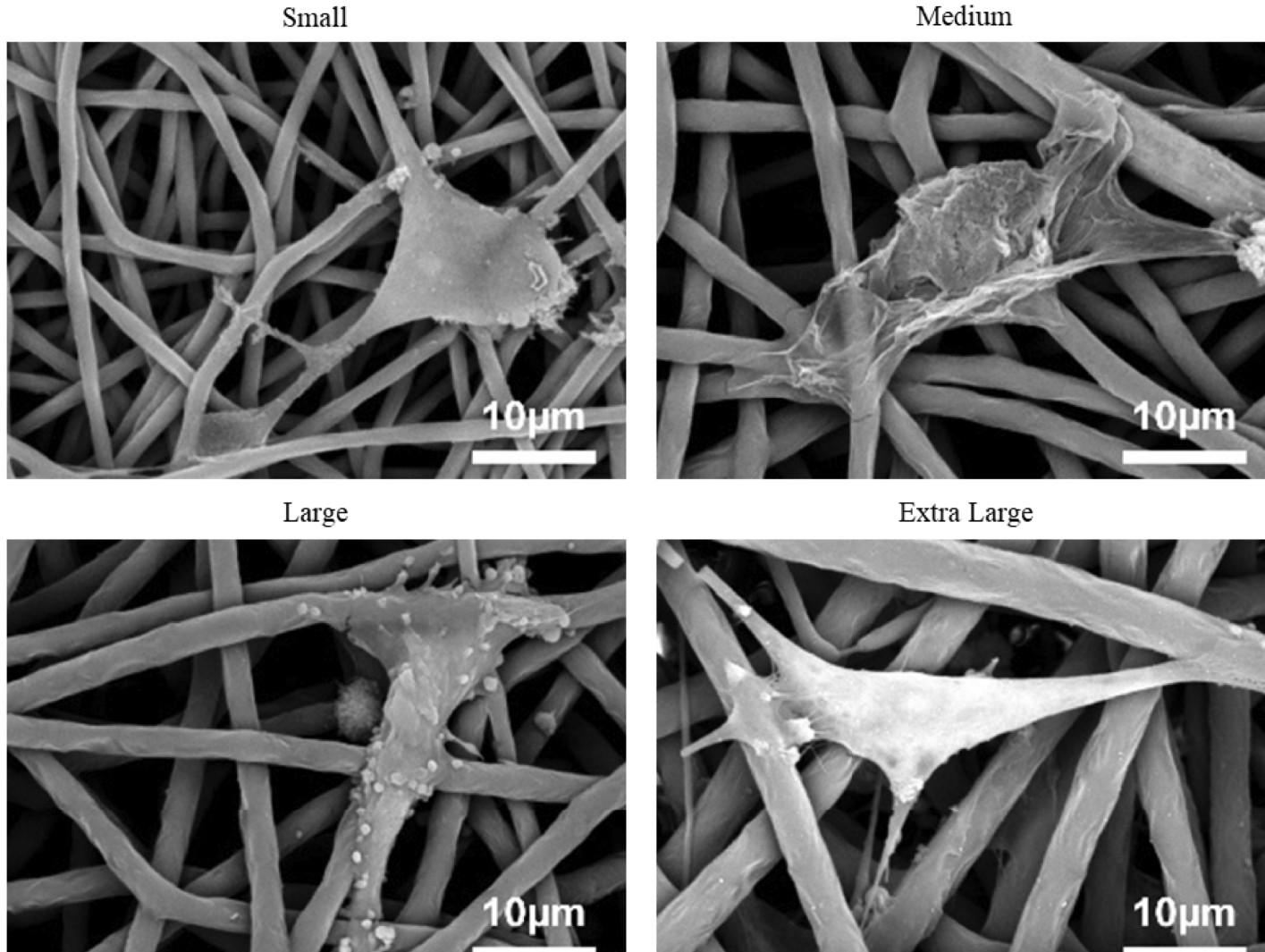

**Fig 3. Representative SEM images of HUVECs binding onto the four scaffold morphologies.**

### Expression of genes associated with hypoxia related apoptosis on PCL and ECM scaffolds cultured in different environments

Gene expression analysis showed some interesting trends (Fig 6D). Firstly, there was a general trend whereby FGF2 expression was lower during hypoxic culture. FGF2 is a key gene for EC angiogenesis [44]. The expression of eNOS showed very little difference between the two scaffolds and the two culture conditions. Likewise, the expression of HIF1-α showed no real trend across the two different scaffolds and two culturing conditions.

### Fluorescence imaging shows cell elongation on ECM scaffolds in normoxic and hypoxic culture environments

Representative DAPI and phalloidin stained cells on both scaffolds under normoxic and hypoxic conditions can be seen in Fig 7. Both scaffolds under both culture conditions showed

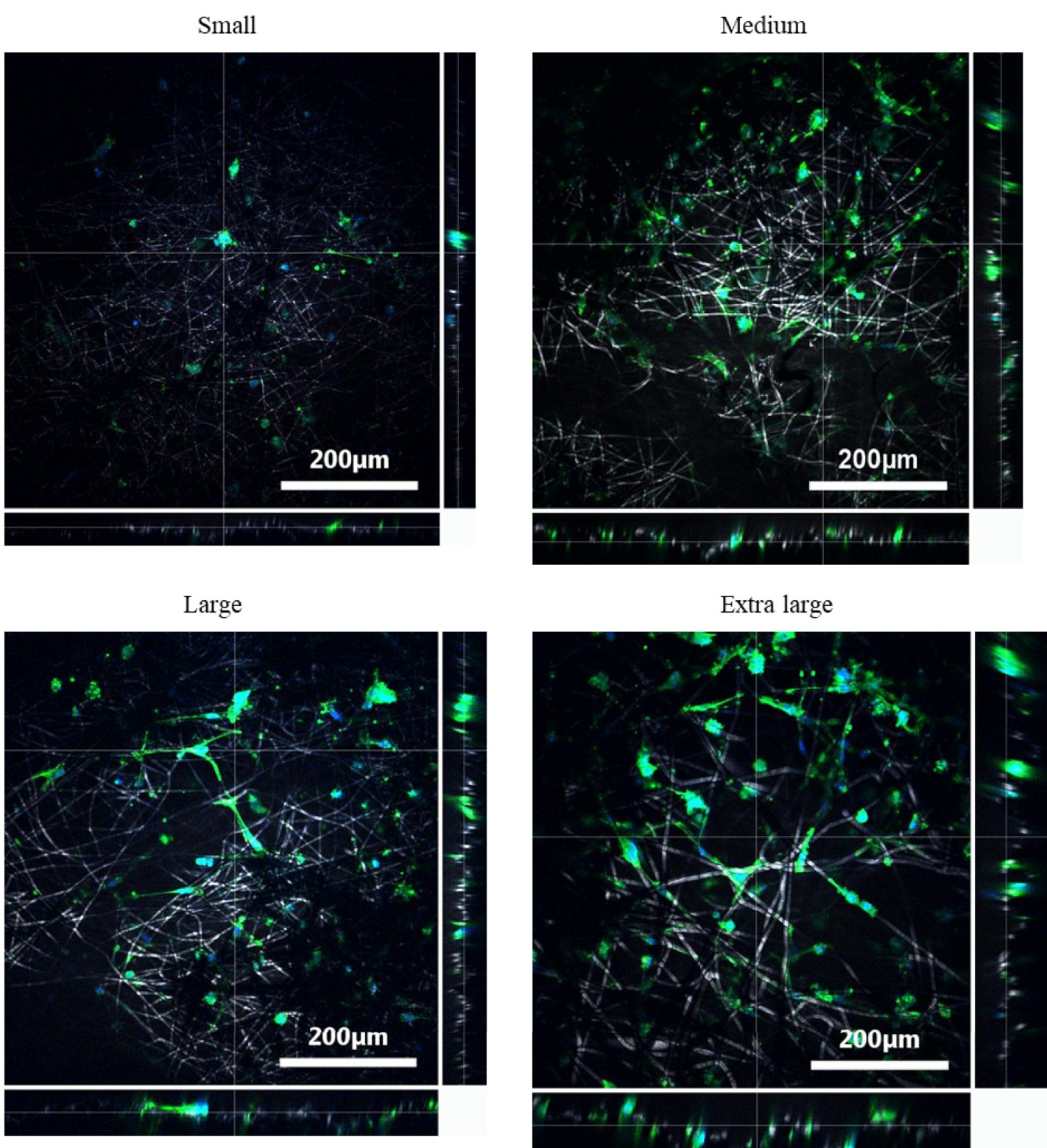

**Fig 4. Representative Z-stack images of DAPI and phalloidin stained HUVECs on all four scaffolds after 12 days of culture.** Images show the depth at which cells can be seen.

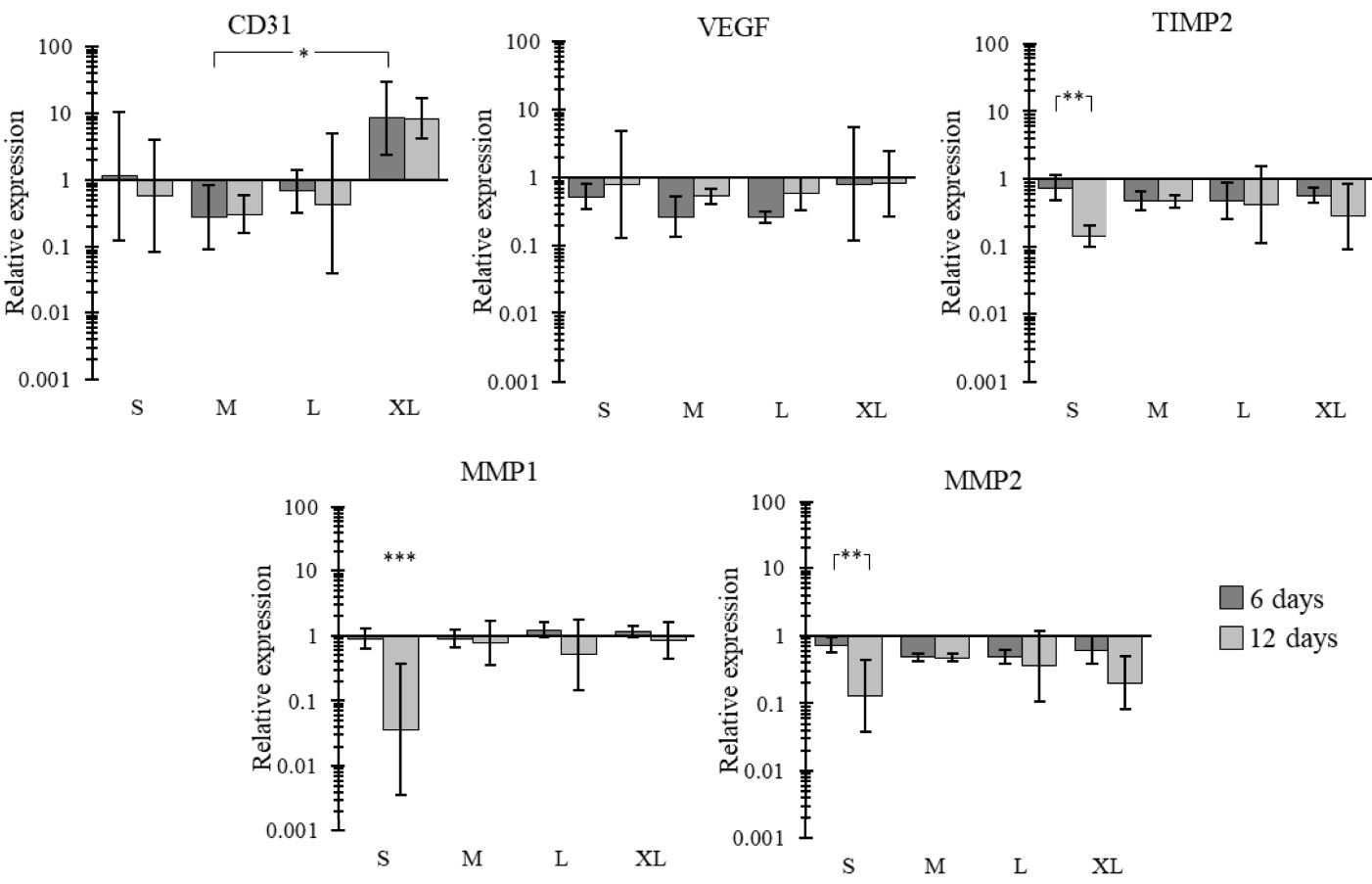

**Fig 5. Gene expression of CD31, VEGF, TIMP2, MMP1 and MMP2 on HUVEC seeded scaffolds for all four scaffolds at 6 days and 12 days relative to 70% confluent HUVECs on tissue culture plastic.** Mean ± SD, n >3.

good cell spreading after 12h of culture. Interestingly, the HUVECs on the ECM scaffold cultured in normoxic conditions after 48h showed more elongation, which is a key phenotypic marker of healthy HUVECs [45].

## Discussion

In this study we have shown that electrospinning offers flexibility as a scaffold manufacturing process, allowing for a range of applications to be pursued using this technique. We were able to create four scaffolds with increasing fibre diameters: 1.64 ± 0.18 μm, 2.95 ± 0.16 μm, 3.37 ± 0.27 μm and 4.83 ± 0.49 μm, with significant differences noted between all four scaffolds. Furthermore, we were also able to create scaffolds that incorporated native decellularized ECM. They were manufactured using PCL, which is an FDA approved material that is both biocompatible and has tailorable bioresorption properties [46, 47].

Tensile testing exposed intriguing results in how fibre diameter altered the mechanical properties of the four scaffolds. It was noted that the stiffness of the small fibre diameter scaffold was significantly lower than that of the three larger fibres for both the 0–5% and 5–10% strain bands (Table 4). Moreover, the larger the fibre diameter the more stiffness it maintained as it was stretched into the 5–10% strain band (Table 4). This particular phenomena may be a

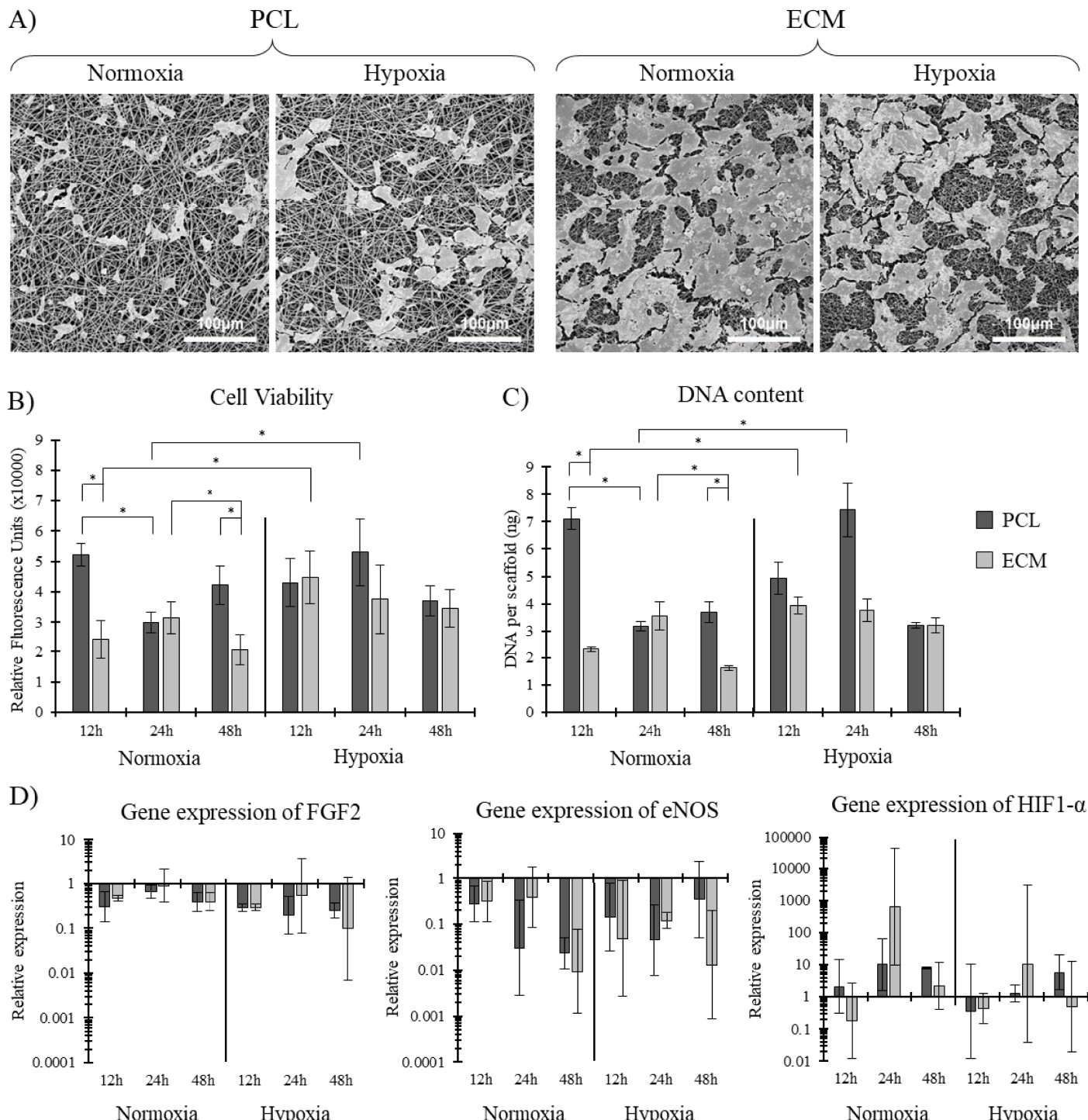

**Fig 6. A)** Representative SEM images of HUVECs cultured on PCL and ECM scaffolds under both normoxic and hypoxic culture conditions after 48h. **B)** Cell viability and **C)** DNA content of HUVECs, n = 4. **D)** Gene expression of three key genes. Mean ± SD, n = 5.

result of the lower porosity, leading to increased bulk properties and therefore merits further research. In addition, all four scaffolds had stiffnesses that were within the range noted in

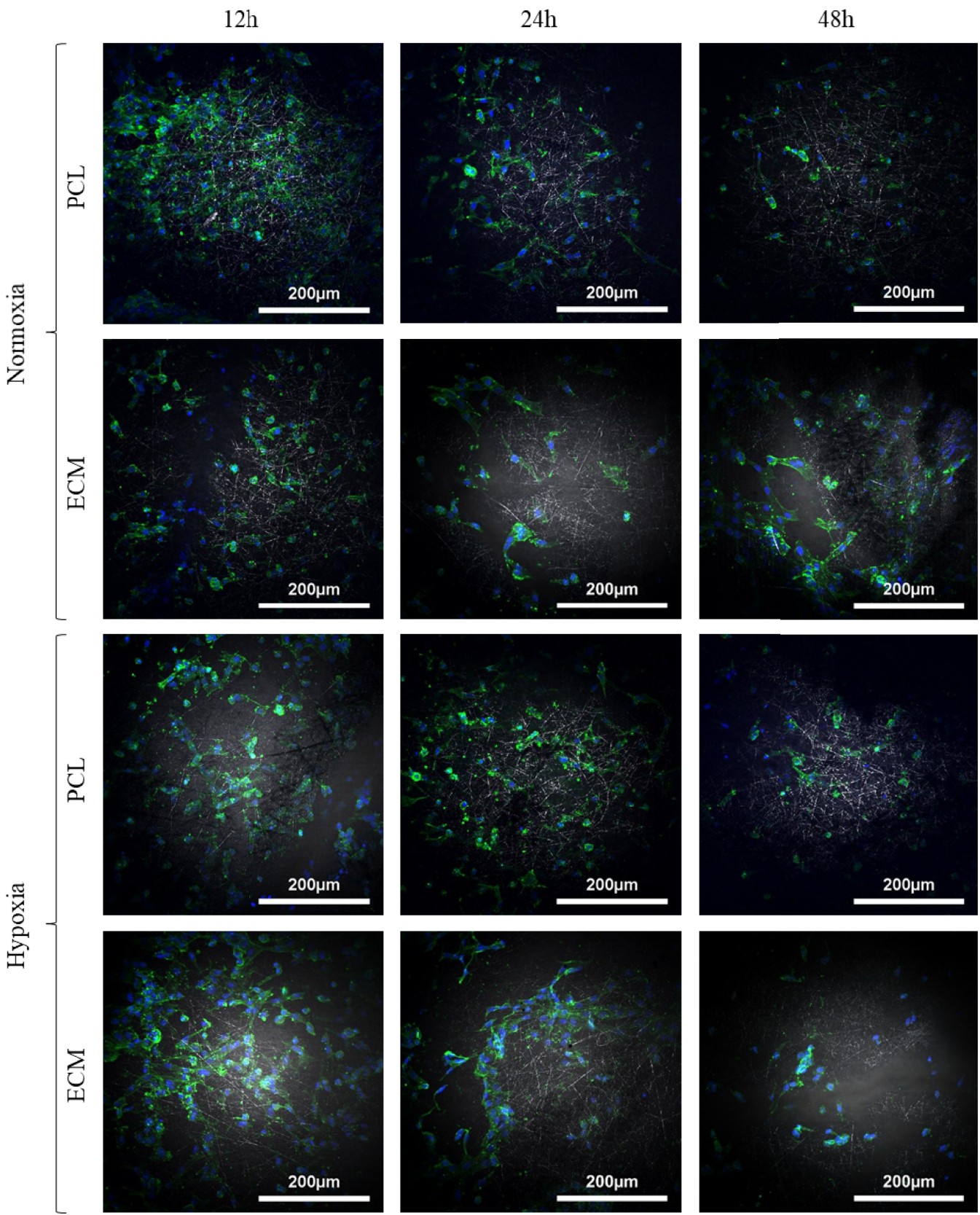

**Fig 7.** Representative DAPI (blue) and phalloidin (green) stained HUVECs on ECM and PCL electrospun scaffolds cultured in hypoxic and normoxic conditions.

native vessels such as the saphenous vein. For example, *Strekelenburg et al.* found that the saphenous vein has longitudinal and circumferential elastic moduli of 23.7 MPa and 4.2 MPa [48]. This circumferential elastic modulus is in a similar realm to the four elastic moduli noted in this study at the 0–5% and 5–10% strains (stiffnesses ranging from 2.17 MPa up to 6.50 MPa depending on strain range and fibre diameter). Previous work by our group also showed how the mechanical properties of a small fibred scaffold could be modulated through the incorporation of ECM [9, 31]. By increasing the quantity of ECM in the fibre, we were able to reduce the Young's Modulus, whilst simultaneously increasing the scaffold's elasticity into the higher strain regions, showing that ECM incorporation can be used as a means of modulating the scaffold's mechanical properties. We matched the fibre diameter between the ECM scaffold and PCL scaffold to ensure that the only variable changing was whether the scaffold contained ECM or not.

Z-stack DAPI and phalloidin fluorescence images (Fig 4) of the scaffolds with varying fibre diameters showed some interesting results. Firstly, there was a clear elongation and spreading of the cell's morphology after 12 days, becoming more obvious with increasing fibre diameter. In contrast to this, the small fibre (and to some extent the medium fibre) appears to show more rounded cells, which may indicate apoptosis [43]. This phenomena is potentially caused by the fibre size and pore size, with evidence shown in other studies examining the effect of electrospun scaffold topography [27, 49, 50]. Secondly, cellular infiltration was noted in the extra-large scaffold after 12 days. Cells were seen at depths of approximately 80 μm, compared to 25 μm for the three other scaffolds. This meant that the HUVEC could grow perpendicular to the top of the scaffold. While this may seem counterintuitive as we are aiming to grown a monolayer of endothelial cells on the surface; this infiltration and 3D cell culture could be helping with cell survival (higher cell viability) and with the quantity of angiogenic genes being actuated (Fig 5). It has been shown in many cases that a 3D cell culture is favourable in relation to gene expression as it more accurately captures the morphological characteristics of the native ECM [51]. This has also been shown when using human coronary artery endothelial cells, where 3D culture showed a significant increase in the expression of key proteins compared to 2D culture [52].

While this study showed that apoptosis was more evident in the HUVECs grown on the smaller fibres, previous pieces of work carried out by our group looking at small fibres (approx. 1μm diameter) have shown that apoptosis can be reduced through the addition of native ECM [9, 31]. These two previous publications looked at how incorporating aortic ECM and cardiac ECM into the PCL fibres altered HUVEC performance. Most notably, results showed that the incorporation of aortic ECM increased DNA content, cell viability and cell proliferation compared to the PCL control and the cardiac ECM scaffold [9]. Furthermore, after 10 days of culture, a significant downregulation of matrix metalloproteinase-1 (MMP1) was noted with the aorta ECM scaffold.

On this note, gene expression results in the fibre diameter study showed some interesting trends. While none of the VEGF gene expression results showed any significance, all four scaffold morphologies showed a relative increase in gene expression between 6 and 12 days, suggesting that VEGF expression is predominantly driven by the cell's maturity, as opposed to scaffold morphology. This trend has been noted before when studying HUVECs. *Zhang, G. et al.* noted a doubling in the relative gene expression of VEGF from HUVECs between 7 and 14 days when cultured on hydroxyapatite scaffolds [53]. The same applies for MMP1, MMP2

and TIMP2, whereby all four morphologies showed a relative decrease in gene expression for these genes after 12 days compared to 6 days. This, like the results for VEGF, suggest that the expression of these genes is mediated more by the cell's maturity than the scaffold's morphology. However, some trends were noted between the four scaffold morphologies. Most notably, all three genes were significantly downregulated in the small fibre scaffold between 6 and 12 days, as opposed to the other morphologies whereby a relative downregulation was still noted, albeit without significance. This suggests that while cell maturity is playing a role in the expression of these genes, the smaller fibre morphology is also having an effect. MMP1 and MMP2 are two genes associated with the remodelling of ECM proteins (collagen and gelatin, respectively). Their downregulation means the cells are not trying to turnover ECM, suggesting homeostasis within the cell microenvironment [54]. Furthermore, an upregulation of these genes is associated with a variety of degenerative diseases [55].

In addition, there was an upregulation of CD31 in the extra-large scaffold compared to the three other scaffolds. CD31 is an angiogenic gene that is heavily involved in endothelial cell-cell association that is required for their reorganization into tubular networks [24]. A significant relative upregulation in CD31 was noted in the extra-large scaffold after 6 days compared to the medium scaffold. Relative increases were also noted between the extra-large scaffold and the three other morphologies at both 6 and 12 days, albeit without significance. No trend in data could be noted between the three other morphologies suggesting that the jump from the large fibre to extra-large fibre (3.37 μm to 4.83 μm) ultimately led to this increase in CD31 expression. While this study was not able to pin point exactly why changing fibre diameter led to alterations in gene expression, it did gives insights into potential mechanisms. For example, Fig 3 shows how the HUVECs are binding to each scaffold, with fewer binding sites present on the largest fibre. A plethora of work has shown how integrin-mediated adhesion in endothelial cells can lead to angiogenesis and other changes in gene expression [56–59]. Therefore, with the final aim of creating an environment most suited for endothelial cells to proliferate into a healthy endothelium, the results suggest that the extra-large scaffold fibre is the best candidate.

Interestingly, eNOS (a central regulator of EC function and endothelial homeostasis) has been shown to downregulate in ECs under hypoxic conditions due to the presence of miR-200b [60, 61]. Our results show very little differences between the scaffolds cultured in normoxia or hypoxia. However, the study by *Janaszak-Jasiecka, A. et al.* found that the majority of this downregulation of the eNOS gene happened in the first 12 h of culture with no more downregulation noted after 16 h of culture [61]. Therefore, the timepoints used in this study may not give an overall image of how the cells are responding in the first 12 h of culture, where the majority of eNOS downregulation has previously been noted. FGF2 is responsible for neovascularization and angiogenesis [44]. Its upregulation is heavily linked with tumour formation [44]. Our results show no significance. Likewise, HIF1-alpha is a gene responsible for apoptosis in ECs when they are subject to hypoxic conditions [62]. Our results found very little difference between the two different culture environments. This may suggest that these electrospun scaffolds possess shielding properties against the upregulation of FGF2 and HIF1-alpha. This may suggest that these scaffolds both slow down angiogenesis and apoptosis in a hypoxic environment.

This study is not without its limitations. Firstly, we were unable to differentiate between whether fibre size or pore size was having the major effect. One issue with electrospinning is that these two morphological characteristics are somewhat interconnected, with a study by *Pham et al.* showing a linear correlation between the two characteristics [63]. The mechanical properties of the scaffolds also changed between each scaffold, meaning that the scaffold's mechanical properties might play a part in the differences in cellular performance noted. A further limitation is that we were unable to study the full spectrum of fibre diameters. We were

only able to look at a limited number of scaffold morphologies and our apparatus only allowed for fibre diameters of up to approximately 5 μm. Furthermore, the system used in this study does not have environmental controls (humidity and temperature), therefore some minor differences in fibre morphology would be expected when repeated. However, the problem of environmental control was kept to a minimum by electrospinning all scaffolds in the same week. In addition, while fibre morphology might change due to environmental factors, the electrospinning parameters used in this study will still create scaffolds with a range of fibre diameters.

More analysis is required in order to get a deeper understanding of the scaffold's topographical properties and how these might affect seeded endothelial cells. One method would be to use technologies such as atomic force microscopy (AFM), which have previously been used to study the interaction of cells with their scaffold/substrate [64, 65]. Furthermore, the infiltration depth of cells could have an impact on the noted gene expression. It has been previously documented that oxygen transfer to deeper parts of a scaffold can be limited depending on the scaffold's architecture [66]. While this is unlikely in the present study, due to the highly porous nature of the scaffolds, it is worth considering in future experiments. Moreover, further analysis on how the scaffolds perform when cultured under flow/shear/pressure/strain would more accurately mimic the native environment. While these limitations merit further study in an attempt to gain an overarching image of exactly how scaffold morphology/composition is affecting the seeded endothelial cells; we have shown that through altering the scaffold's fibre diameter that we can influence cellular performance.

## Conclusions

In this study, three different aspects of the scaffold and culture environment were assessed to evaluate which had the biggest impact on HUVEC performance. It was noted that the incorporation of ECM into the fibres had minimal effects on HUVEC gene expression when cultured in both a hypoxic and normoxic environment, but it did have the effect of increasing cell deposits on the scaffold. Furthermore, the hypoxic culture had a limited effect on the HUVECs when cultured on both the PCL and the ECM scaffolds.

On the other hand, fibre diameter had the greatest influence on cell performance. The largest fibre diameter led to increased HUVEC infiltration and significantly increased cell viability and expression of CD31 compared to the three other fibre diameters (significance was only noted between the extra-large fibre and medium fibre at 6 days of culture).

Therefore, the present study suggests that fibre diameter plays the biggest role in the modulation of cell proliferation and angiogenic gene response.

## Supporting information

**S1 Fig. Cell viability of HUVECs grown on tissue culture plastic (positive control) compared to HUVECs grown on a small fibred PCL scaffold after 4 days of culture.** (N = 4). (EPS)

**S1 Data.**
(ZIP)

## Acknowledgments

The authors would like to thanks Prof. Alistair Elfick for use of lab facilities (Institute of Bioengineering, the University of Edinburgh).

## Author Contributions

**Conceptualization:** James Alexander Reid, Anthony Callanan.

**Data curation:** James Alexander Reid, Alison McDonald.

**Formal analysis:** James Alexander Reid.

**Funding acquisition:** James Alexander Reid, Alison McDonald, Anthony Callanan.

**Investigation:** James Alexander Reid.

**Methodology:** James Alexander Reid.

**Project administration:** James Alexander Reid.

**Resources:** James Alexander Reid, Anthony Callanan.

**Software:** James Alexander Reid, Alison McDonald.

**Supervision:** Anthony Callanan.

**Validation:** James Alexander Reid.

**Visualization:** James Alexander Reid.

**Writing – original draft:** James Alexander Reid.

**Writing – review & editing:** James Alexander Reid.

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
