## [Decision Letter · Decision Letter 0]

30 Jul 2020

PONE-D-20-16177

Modulating Electrospun Polycaprolactone Scaffold Morphology and Composition to Alter Endothelial Cell Proliferation and Angiogenic Gene Response

PLOS ONE

Dear Dr. Callanan,

Thank you for submitting your manuscript to PLOS ONE. After careful consideration, we feel that it has merit but does not fully meet PLOS ONE’s publication criteria as it currently stands. Therefore, we invite you to submit a revised version of the manuscript that addresses the points raised during the review process.

We look forward to receiving your revised manuscript.

Kind regards,

Feng Zhao

Academic Editor

PLOS ONE

Journal Requirements:

Reviewer's Responses to Questions

**Comments to the Author**

1. Is the manuscript technically sound, and do the data support the conclusions?

Reviewer #1: Partly

Reviewer #2: No

2. Has the statistical analysis been performed appropriately and rigorously? 

Reviewer #1: No

Reviewer #2: Yes

3. Have the authors made all data underlying the findings in their manuscript fully available?

Reviewer #1: Yes

Reviewer #2: Yes

4. Is the manuscript presented in an intelligible fashion and written in standard English?

Reviewer #1: Yes

Reviewer #2: Yes

5. Review Comments to the Author

Reviewer #1: The paper is well written and logically ordered. The authors need to focus on choosing either fiber diameter or hypoxia and delve deeper into its dependence on endothelial cell behavior. Please refer to the attachment for the reviewer comments for detailed comments.

Reviewer #2: Comments:

The research topic on electrospun fiber scaffold for tissue engineering is promising. However, the experimental design of this work will produce very limited new information in the field. The motivation of this research is not clear.

1. Why did authors tried to compare the ECs gene expression under hypoxic normoxia or hypoxia condition?

2. What is the biological mechanism for increase in viability and cell proliferation on the large diameter fiber scaffold?

---

## [Author Response · Author response to Decision Letter 0]

31 Aug 2020

Response to Reviewers:

Reviewer 1:

Overall Comment:

The paper is well written and logically ordered. The authors need to focus on choosing either fiber diameter or hypoxia and delve deeper into its dependence on endothelial cell behavior. Please refer to the attachment for the reviewer comments for detailed comments.

Response:

The authors would like to thank you for taking the time in reviewing the manuscript. While we agree that choosing a focus is important, we believe that all the conditions we have studied are important to the study. Our conclusion is that the major finding of this study is that fibre diameter has the biggest effect on endothelial cell behaviour. We have therefore used the majority of the discussion to delve deeper into fibre diameter and put less of a focus on the other factors we studied. We have altered the structure and focus our Conclusions to ensure this comes across as our major finding/focus.

Edits made to Conclusions section:

In this study, three different aspects of the scaffold and culture environment were assessed to evaluate which had the biggest impact on HUVEC performance. It was noted that the incorporation of ECM into the fibres had minimal effects on HUVEC gene expression when cultured in both a hypoxic and normoxic environment, but it did have the effect of increasing cell deposits on the scaffold. Furthermore, the hypoxic culture had a limited effect on the HUVECs when cultured on both the PCL and the ECM scaffolds.

On the other hand, fibre diameter had the greatest influence on cell performance. The largest fibre diameter led to increased HUVEC infiltration and significantly increased cell viability and expression of CD31 compared to the three other fibre diameters (significance was only noted between the extra-large fibre and medium fibre at 6 days of culture). 

Therefore, the present study suggests that fibre diameter plays the biggest role in the modulation of cell proliferation and angiogenic gene response.

Comment 1:

The author states in the introduction that ‘fibre diameter on endothelial cells has not yet been studied’. However, there has been ample research done on the effects of fiber diameter on endothelial cell behavior. 

Some of the references are

 Rüder, Constantin, Tilman Sauter, Karl Kratz, Tobias Haase, Jan Peter, Friedrich Jung, Andreas Lendlein, and Dietlind Zohlnhöfer. "Influence of fibre diameter and orientation of electrospun copolyetheresterurethanes on smooth muscle and endothelial cell behaviour." Clinical hemorheology and microcirculation 55, no. 4 (2013): 513-522.

 Ko, Young-Gwang, Ju Hee Park, Jae Baek Lee, Hwan Hee Oh, Won Ho Park, Donghwan Cho, and Oh Hyeong Kwon. "Growth behavior of endothelial cells according to electrospun poly (D, L-lactic-co-glycolic acid) fiber diameter as a tissue engineering scaffold." Tissue engineering and regenerative medicine 13, no. 4 (2016): 343-351.

The authors need to cite appropriate references and remove claims that are not justified. 

Response:

Thank you very much for comment and for finding those references. All references will be included and any unjustified statement will be removed. Statements will also be amended to emphasise that this is the first time (to the best of our knowledge) that the effect of fibre diameter using polycaprolactone will be assessed with endothelial cells.

Edits made to Introduction section:

For example, it has been shown with human kidney primary epithelial cells and chondrocytes that fibre diameter and fibre orientation have dramatic effects on cell morphology and the expression of key genes [27,28]. While there has been some work looking at fibre diameter and endothelial cells, to the best of our knowledge, the present study is the first that looks at the effect of fibre diameter in electrospun PCL scaffolds on endothelial cells [29,30]. Furthermore, recent work by our group has shown that incorporating native vascular ECMs into the fibre had positive effects on HUVEC proliferation and gene expression [9,31].

Comment 2:

The authors need to check the paper for typographical errors and grammatical mistakes. Eg. Superscript -1 in the sentence ‘Coherent anti-stokes Raman scattering (CARS) imaging at 2911cm-1 was used to image the PCL scaffold fibres, whilst simultaneously exciting two photon fluorescence (TPEF) from Phalloidin and DAPI stained cells’.

Response:

Thank you for the comment and picking up on the mistake. All mistakes like this will be changed in the text and the paper will be thoroughly proofread.

Comment 3:

The authors need to clarify on how the diameters of fibers were measured. How many FESEM images for each sample were used to obtain the statistical significance with regard to fiber diameter? The difference between the 3um and 4um fibers are quite close. Is it possible that the diameters might have not been statistically significant if other images of the same sample were used?

Response:

Thank you for the comment regarding more detail on how the fibres were measured. Fibre diameters were measured using a minimum of 50 individual fibres from four different scaffolds. Due to fibre morphology being fairly uniform throughout the scaffold, a small standard deviation was observed. Furthermore due to the high number of measurements taken and the small standard deviation, the differences noted between the 3um and 4um were significantly different. Information will be added to the methods section to clarify how fibre diameter is measured.

Edits made to Methods section:

 Fibre and pore properties

Scanning electron images were analysed using ImageJ software (NIH). Briefly, SEM images of the scaffolds analysed using the DiameterJ plugin for fibre diameter and pore width and the OrientationJ plugin for fibre orientation [33]. Fibre diameters were measured on a minimum for 50 different fibres on four different scaffolds. Variance in fibre diameters along individual fibres was measured using 5 diameter measurements per fibre along a total of 5 fibres.

Comment 4:

The authors need to elaborate on the reproducibility of the electrospinning results. Environmental factors play a major role in the diameter and the subsequent fiber morphology.

Response:

Thank you for the comment. You are absolutely correct that environmental factors do play a major role. Our system is not environmentally controlled and work has shown that temperature and humidity both affect electrospinning, so this will be added as a limitation into the discussion [1]. We electrospun all the scaffolds in the same week, which would limit the overall effect that the environment has between each scaffold. While these electrospinning parameters would not produce identical scaffolds when reproduced, they would lead to similar scaffolds with differing fibre diameters. 

Edits made to Discussion section:

This study is not without its limitations. Firstly, we were unable to differentiate between whether fibre size or pore size was having the major effect. One issue with electrospinning is that these two morphological characteristics are somewhat interconnected, with a study by Pham et al. showing a linear correlation between the two characteristics [59]. A further limitation is that we were unable to study the full spectrum of fibre diameters. We were only able to look at a limited number of scaffold morphologies and our apparatus only allowed for fibre diameters of up to approximately 5μm. Furthermore, the system used in this study does not have environmental controls (humidity and temperature), therefore some minor differences in fibre morphology would be expected when repeated. However, the problem of environmental control was kept to a minimum by electrospinning all scaffolds in the same week. In addition, while fibre morphology might change due to environmental factors, the electrospinning parameters used in this study will still create scaffolds with a range of fibre diameters. Moreover, further analysis on how the scaffolds perform when cultured under flow/shear would more accurately mimic the native environment. While these limitations merit further study in an attempt to gain an overarching image of exactly how scaffold morphology/composition is affecting the seeded endothelial cells; we have shown that through altering the scaffold’s fibre diameter that we can influence cellular performance.

Comment 5:

The authors have performed a one-way ANOVA. Why did the authors choose ANOVA and not regression-based analysis? 

Response:

Thank you very much for the comment. One-way ANOVA with tukey post hoc was used as this is the most commonly used method for determining statistical significance between means of three or more independent groups. This method of measuring significance has been widely used in similar biological analyses [2–8]. (reference list can be found at the end of this document)

Comment 6:

The mechanical properties and stiffness of the fibers also changes along with diameter. Why have the authors chosen to just represent that fiber diameter is the sole factor leading to changes in expression? The authors need to think about performing AFM if possible to get a better understanding about the scaffold properties. 

Response:

Thank you for the comment. While the mechanical properties and stiffness of the fibres are changing between the scaffolds, the differences noted are small on the grand scheme of stiffnesses that have been tested in vitro. You are correct that the assumption cannot be made, therefore a comment will be added to the limitations to discuss this point. With regards to performing AFM, this would give us a better understanding of the scaffold’s topographical properties, however this equipment is not available to us. A discussion point will be added to discuss how AFM could be utilised in future studies.

Edits made to Discussion section:

One issue with electrospinning is that these two morphological characteristics are somewhat interconnected, with a study by Pham et al. showing a linear correlation between the two characteristics [59]. The mechanical properties of the scaffolds also changed between each scaffold, meaning that the scaffold’s mechanical properties might play a part in the differences in cellular performance noted. A further limitation is that we were unable to study the full spectrum of fibre diameters. We were only able to look at a limited number of scaffold morphologies and our apparatus only allowed for fibre diameters of up to approximately 5μm.

More analysis is required in order to get a deeper understanding of the scaffold’s topographical properties and how these might affect seeded endothelial cells. One method would be to use technologies such as atomic force microscopy (AFM), which have previously been used to study the interaction of cells with their scaffold/substrate [60.61]. Moreover, further analysis on how the scaffolds perform when cultured under flow/shear would more accurately mimic the native environment. While these limitations merit further study in an attempt to gain an overarching image of exactly how scaffold morphology/composition is affecting the seeded endothelial cells; we have shown that through altering the scaffold’s fibre diameter that we can influence cellular performance.

Comment 7:

The thickness of the fabricated scaffolds needs to be mentioned in the main paper. Differences in cell infiltration can only be compared when the thickness of the scaffolds are similar. The variable thickness could also be a factor in expression of genes. The author needs to further investigate and elaborate on the different variables. 

Response:

Thank you for the comments. The thickness of the scaffold will be added to the paper. While we agree that thickness can have an impact on infiltration, we have only noted infiltrations that are in the top 25-80um of the scaffold. These infiltration depths will be the same whether the scaffold is 300um or 3000um thick. All the scaffolds used in this study were much thicker than the deepest noted infiltration, therefore thickness should not have had an effect on infiltration. The same applies for gene expression whereby the cells will not be affected by any excess thickness. However, thickness would play a major role in nutrient delivery, especially when cells were found deeper in the scaffold, which would cascade down into gene expression. However, due to infiltration depths being less than 100um, this should not have an effect on in vitro cultured cells.

Edits made to Results section:

Table 4: Mechanical and physical properties of all four scaffolds.

 Fibre diameter (µm) Pore diameter (μm) Variance in fibre diameter along fibre (%) Scaffold thickness (µm) Porosity (%) Ultimate tensile strength (MPa) Failure strain (%) Contact angle at 0.2s (°) Stiffness (MPa)

 0-5% 5-10%

S 1.64 ± 0.18 8.4 ± 3.7 1.23 140 ± 9.4 91.0 ± 1.6 1.08 ± 0.17 843 ± 143 110.8 ± 7.7 4.10 ± 0.33 2.17 ± 0.25

M 2.95 ± 0.16 14.7 ± 4.8 0.83 336 ± 36.9 89.6 ± 2.4 1.50 ± 0.09 1202 ± 62 128.2 ± 4.0 6.50 ± 0.72 2.99 ± 0.46

L 3.37 ± 0.27 16.4 ± 6.7 2.28 549 ± 33.8 83.9 ± 1.1 1.55 ± 0.16 1071 ± 21 129.1 ± 2.5 5.80 ± 0.36 3.16 ± 0.48

XL 4.83 ± 0.49 23.3 ± 9.0 2.83 598 ± 59.6 83.3 ± 0.9 1.26 ± 0.06 855 ± 120 131.5 ± 1.1 5.77 ± 0.39 3.66 ± 0.28

Edits made to Discussion section:

Furthermore, the infiltration depth of cells could have an impact on the noted gene expression. It has been previously documented that oxygen transfer to deeper parts of a scaffold can be limited depending on the scaffold’s architecture [62]. While this is unlikely in the present study due to the highly porous nature of the scaffolds, it is worth considering in future experiments. Moreover, further analysis on how the scaffolds perform when cultured under flow/shear would more accurately mimic the native environment. While these limitations merit further study in an attempt to gain an overarching image of exactly how scaffold morphology/composition is affecting the seeded endothelial cells; we have shown that through altering the scaffold’s fibre diameter that we can influence cellular performance.

Comment 8:

There are no appropriate controls for the cell titer blue viability assays. A positive and negative control needs to be represented. A standard tissue culture is an ideal positive control. 

Response:

Negative controls were run alongside and they are used to remove the background fluorescence from the data. All data has been represented with this background fluorescence removed. Any value above 0 can be counted as excess fluorescence over the background. A positive control of cells grown on tissue culture plastic can be added, however, this study is looking at comparisons between scaffolds and not making comparisons with a standard culture on tissue culture plastic. We have added a supplementary file showing a comparison between cells cultured on tissue culture plastic and those cultured on scaffolds.

Edits made to Methods section:

 CellTiter-Blue® cell viability assay

The assay was performed as per manufacturer’s instructions (Promega) and as described in previous studies[9]. Measurements were taken after 3.5 h at Ex: 525 nm and Em: 580-640 nm. For each condition group, n=4. All data has been represented with the background fluorescence removed (negative control). A comparison of cells cultured on scaffolds to cells cultured on tissue culture plastic can be found in S1 Fig.

Edits made to Supplementary section:

Caption: Cell viability of HUVECs grown on tissue culture plastic (positive control) compared to HUVECs grown on a small fibred PCL scaffold after 4 days of culture. (N = 4).

Comment 9:

Controls need to be established for the gene expression studies.

Response:

Thank you very much for the comment. HUVECs were cultured on tissue culture plastic up until 70% confluence. These cells were then run for all the genes and are used in the ΔΔCt method. All values in the gene expression graphs were normalized to the values found for HUVECs on tissue culture plastic. Therefore, the positive control is represented as 1 on each graph and all other values are relative to this positive control. Negative controls were run to ensure no contamination but were not included in the graph as they did not show up values when analysed.

Edits made to Methods section:

 Reverse transcription polymerase chain reaction (RT-PCR)

RNA was extracted from the cell seeded scaffolds using a Tri-Reagent (Invitrogen, Thermofisher) method and purified using Qiagen’s RNeasy spin column system. Real-time polymerase chain reaction was performed using a LightCycler® 480 Instrument II (Roche Life Science) and Sensifast™ SYBR® High-ROX system (Bioline). Forward and reverse sequences were designed online. Relative quantification of RT-PCR results was carried out using the 2^(-∆∆ct) method [41]. Gene expression levels were expressed relative to GAPDH (housekeeping gene) and normalised to 70% confluent HUVECs on tissue culture plastic (positive control). 

Comment 10:

Surface characterization of the fibers need to be performed in order to demonstrate that the surface properties are similar across all scaffold morphologies. 

Response:

Thank you for the comment. While we agree that more advanced scaffold analysis would be helpful in comparing the scaffolds, however, this is not in the scope of this submitted paper. The highest resolution images our SEM was capable of taking showed no difference between the different fibres. While SEM images don’t show the nanotopography of the fibres, they do show that all fibres are uniform in size (as seen with the small standard deviation) and they also show that the fibres all appear smooth in structure and no melting of fibres is occurring. 

We have performed some analysis on the SEM images to show the variability in fibre diameter along each individual fibre.

Edits made to Methods section:

 Fibre and pore properties

Scanning electron images were analysed using ImageJ software (NIH). Briefly, SEM images of the scaffolds analysed using the DiameterJ plugin for fibre diameter and pore width and the OrientationJ plugin for fibre orientation [33]. Fibre diameters were measured on a minimum for 50 different fibres on four different scaffolds. Variance in fibre diameters along individual fibres was measured using 5 diameter measurements per fibre along a total of 5 fibres.

Edits made to Results section:

Table 4: Mechanical and physical properties of all four scaffolds.

 Fibre diameter (µm) Pore diameter (μm) Variance in fibre diameter along fibre (%) Scaffold thickness (µm) Porosity (%) Ultimate tensile strength (MPa) Failure strain (%) Contact angle at 0.2s (°) Stiffness (MPa)

 0-5% 5-10%

S 1.64 ± 0.18 8.4 ± 3.7 1.23 140 ± 9.4 91.0 ± 1.6 1.08 ± 0.17 843 ± 143 110.8 ± 7.7 4.10 ± 0.33 2.17 ± 0.25

M 2.95 ± 0.16 14.7 ± 4.8 0.83 336 ± 36.9 89.6 ± 2.4 1.50 ± 0.09 1202 ± 62 128.2 ± 4.0 6.50 ± 0.72 2.99 ± 0.46

L 3.37 ± 0.27 16.4 ± 6.7 2.28 549 ± 33.8 83.9 ± 1.1 1.55 ± 0.16 1071 ± 21 129.1 ± 2.5 5.80 ± 0.36 3.16 ± 0.48

XL 4.83 ± 0.49 23.3 ± 9.0 2.83 598 ± 59.6 83.3 ± 0.9 1.26 ± 0.06 855 ± 120 131.5 ± 1.1 5.77 ± 0.39 3.66 ± 0.28

 

Reviewer 2:

Overall Comment:

The research topic on electrospun fiber scaffold for tissue engineering is promising. However, the experimental design of this work will produce very limited new information in the field. The motivation of this research is not clear.

Response:

The authors would like to thank you for taking the time in reviewing the manuscript. We agree that the motivation may not be that clear. We have rewritten some of the introduction to help clarify the motivation of this paper.

Edits made to Introduction section:

While these ECM scaffolds have been looked at in normoxic culture, they have not yet been studied in hypoxic cultures that more accurately mimic the native oxygen content found in most native tissue types. Therefore, there is further motivation in not only studying fibre diameter but also looking at how hypoxic culture affects blended PCL/ECM electrospun scaffolds. By looking at these different compositional and morphological aspects of the electrospun scaffold, the aim is to pin-point which one has the biggest effect on seeded endothelial cells to help guide future scaffold design.

Comment 1:

Why did authors tried to compare the ECs gene expression under hypoxic normoxia or hypoxia condition? 

Response:

Thank you very much for the comment. The reason we have compared normal incubator conditions (normoxia) to hypoxic conditions is that hypoxia is a truer representation of the in vivo conditions. Most in vitro experiments take place in 16% oxygen which is higher than the oxygen content found in most parts of the body. For example, most healthy tissues in the body have oxygen contents ranging between 3% and 7.4%, and most tumours have oxygen contents ranging between 0.3% and 4.2% [9].

Comment 2:

What is the biological mechanism for increase in viability and cell proliferation on the large diameter fiber scaffold?

Response:

Thank you for the comment. We will try and discuss this further in the discussion. While we don’t know the exact answer to this, one theory is that the changes in viability/gene expression are being driven by the integrin binding sites which have previously been shown to drive gene expression in endothelial cells [10–13].

Edits made to Discussion section:

In addition, there was an upregulation of CD31 in the extra-large scaffold compared to the three other scaffolds. CD31 is an angiogenic gene that is heavily involved in endothelial cell-cell association that is required for their reorganization into tubular networks [24]. A significant relative upregulation in CD31 was noted in the extra-large scaffold after 6 days compared to the medium scaffold. Relative increases were also noted between the extra-large scaffold and the three other morphologies at both 6 and 12 days, albeit without significance. No trend in data could be noted between the three other morphologies suggesting that the jump from the large fibre to extra-large fibre (3.37µm to 4.83µm) ultimately led to this increase in CD31 expression. While this study was not able to pin point exactly why changing fibre diameter led to alterations in gene expression, it did gives insights into potential mechanisms. For example, Figure 3 shows how the HUVECs are binding to each scaffold, with fewer binding sites present on the largest fibre. A plethora of work has shown how integrin-mediated adhesion in endothelial cells can lead to angiogenesis and other changes in gene expression [56-59]. Therefore, with the final aim of creating an environment most suited for endothelial cells to proliferate into a healthy endothelium, the results suggest that the extra-large scaffold fibre is the best candidate. 

References

1. De Vrieze S, Van Camp T, Nelvig A, Hagström B, Westbroek P, De Clerck K. The effect of temperature and humidity on electrospinning. J Mater Sci. 2009;44: 1357–1362. doi:10.1007/s10853-008-3010-6

2. Elliott MB, Ginn B, Fukunishi T, Bedja D, Suresh A, Chen T, et al. Regenerative and durable small-diameter graft as an arterial conduit. Proc Natl Acad Sci U S A. 2019;116: 12710–12719. doi:10.1073/pnas.1905966116

3. Hansen KJ, Laflamme MA, Gaudette GR. Development of a Contractile Cardiac Fiber From Pluripotent Stem Cell Derived Cardiomyocytes. Front Cardiovasc Med. 2018;5: 1–11. doi:10.3389/fcvm.2018.00052

4. Young RE, Graf J, Miserocchi I, Van Horn RM, Gordon MB, Anderson CR, et al. Optimizing the alignment of thermoresponsive poly(N-isopropyl acrylamide) electrospun nanofibers for tissue engineering applications: A factorial design of experiments approach. PLoS One. 2019;14: 1–15. doi:10.1371/journal.pone.0219254

5. Munir N, Larsen RS, Callanan A. Fabrication of 3D cryo-printed scaffolds using low-temperature deposition manufacturing for cartilage tissue engineering. Bioprinting. 2018;10: 1–8. doi:10.1016/j.bprint.2018.e00033

6. Raphel L, Talasila A, Cheung C, Sinha S. Myocardin Overexpression Is Sufficient for Promoting the Development of a Mature Smooth Muscle Cell-Like Phenotype from Human Embryonic Stem Cells. PLoS One. 2012;7. doi:10.1371/journal.pone.0044052

7. Allen ACB, Barone E, Momtahan N, Crosby CO, Tu C, Deng W, et al. Temporal Impact of Substrate Anisotropy on Differentiating Cardiomyocyte Alignment and Functionality. Tissue Eng - Part A. 2019;25: 1426–1437. doi:10.1089/ten.tea.2018.0258

8. Delaine-Smith RM, Green NH, Matcher SJ, MacNeil S, Reilly GC. Monitoring fibrous scaffold guidance of three-dimensional collagen organisation using minimally-invasive second harmonic generation. PLoS One. 2014;9. doi:10.1371/journal.pone.0089761

9. McKeown SR. Defining normoxia, physoxia and hypoxia in tumours - Implications for treatment response. Br J Radiol. 2014;87: 1–12. doi:10.1259/bjr.20130676

10. Brizzi MF, Defilippi P, Rosso A, Venturino M, Garbarino G, Miyajima AG, et al. Integrin-mediated adhesion of endothelial cells induces JAK2 and STAT5A activation: Role in the control of c-fos gene expression. Mol Biol Cell. 1999;10: 3463–3471. doi:10.1091/mbc.10.10.3463

11. Malinin NL, Pluskota E, Byzova T V. Integrin signaling in vascular function. Curr Opin Hematol. 2012;19: 206–211. doi:10.1097/MOH.0b013e3283523df0

12. Post A, Wang E, Cosgriff-Hernandez E. A Review of Integrin-Mediated Endothelial Cell Phenotype in the Design of Cardiovascular Devices. Ann Biomed Eng. 2019;47: 366–380. doi:10.1007/s10439-018-02171-3

13. Dejana E, Raiteri M, Resnati M, Lampugnani MG. Endothelial integrins and their role in maintaining the integrity of the vessel wall. Kidney Int. 1993;43: 61–65. doi:10.1038/ki.1993.11

---

## [Decision Letter · Decision Letter 1]

24 Sep 2020

Modulating Electrospun Polycaprolactone Scaffold Morphology and Composition to Alter Endothelial Cell Proliferation and Angiogenic Gene Response

PONE-D-20-16177R1

Dear Dr. Callanan,

We’re pleased to inform you that your manuscript has been judged scientifically suitable for publication and will be formally accepted for publication once it meets all outstanding technical requirements.

Kind regards,

Feng Zhao

Academic Editor

PLOS ONE

---

## [Editor Report · Acceptance letter]

28 Sep 2020

PONE-D-20-16177R1 

Modulating electrospun polycaprolactone scaffold morphology and composition to alter endothelial cell proliferation and angiogenic gene response 

Dear Dr. Callanan:

I'm pleased to inform you that your manuscript has been deemed suitable for publication in PLOS ONE. Congratulations! Your manuscript is now with our production department. 

Kind regards, 

on behalf of

Dr. Feng Zhao 

Academic Editor

PLOS ONE